Malignant behaviors and immune response in melanoma: Epstein-Barr virus induced gene 3 as a therapeutic target based on an in-vitro exploration

Zhang Ying 1
Cheng Fengrui 1
Cai Xingrui 2
Wu Jingping 1 jingpingwu2022@163.com
1 Department of Medical Aesthetics, Affiliated Hospital of Chengdu University of Traditional Chinese Medicine , Chengdu , China
2 Surgery of Traditional Chinese Medicine, School of Clinical Medicine, Chengdu University of Traditional Chinese Medicine , Chengdu , China
Wang Jincheng
Electronic publication date: 2024 Dec 23
Publication date: 2024
Volume: 12
Electronic Location ID: e18730
Received 2024 Oct 10; Accepted 2024 Nov 27
Copyright: © 2024 Zhang et al.
Copyright year: 2024
Copyright holder: Zhang et al.
License: This is an open access article distributed under the terms of the Creative Commons Attribution License, which permits unrestricted use, distribution, reproduction and adaptation in any medium and for any purpose provided that it is properly attributed. For attribution, the original author(s), title, publication source (PeerJ) and either DOI or URL of the article must be cited.
License URL: https://creativecommons.org/licenses/by/4.0/

Keywords: Malignant behaviors, Immune response, Melanoma, Macrophage, Epstein-Barr Virus Induced Gene 3

Funding: Sanhuang Immunity Enhancing Soup Combined with Dual Immunotherapy in the Treatment of Malignant Melanoma 2023zd019 This study was supported by the Mechanism of sensitisation and clinical validation study of Sanhuang Immunity Enhancing Soup combined with dual immunotherapy in the treatment of malignant melanoma (2023zd019). The funders had no role in study design, data collection and analysis, decision to publish, or preparation of the manuscript.

==============================
Background

Epstein-Barr virus induced gene 3 (EBI3), a member of the IL-12 family, is known to be involved in malignant progression in a variety of cancers, but its role in melanoma is unclear. The aim of this study was to explore the effects of EBI3 on the malignant phenotype melanoma to reveal its potential as a therapeutic target.

Methods

In this study, we used bioinformatics to analyze the expression of EBI3 in pan-cancer and verified its expression level in melanoma cells by reverse transcription-quantitative polymerase chain reaction (RT-qPCR). Subsequently, the effects of EBI3 knockdown on cell proliferation, migration and invasion were detected using the Cell Counting Kit-8 (CCK-8) and Transwell assays. Changes in immune-related cytokines were detected by ELISA, and macrophage polarization was observed using immunofluorescence. Finally, the phosphorylation levels of signaling pathways such as Smad3, STAT6 and cGAS-STING were analyzed by Western blot.

Results

EBI3 was evidently highly-expressed in melanoma, and silencing of EBI3 could visibly suppress the survival and migration/invasion of melanoma cells, concurrent with the increased levels of BAX and CDH1 and the decreased expressions of BCL2 and CDH2. Meanwhile, EBI3 knockdown diminished the phosphorylation levels of both Smad3 and STAT6 and the levels of immune response-relevant cytokines in melanoma cells, while aggravating the macrophage M1 polarization and the expression of cGAS, p-STING and p-IRE1 α in THP-1 monocyte-derived macrophages co-cultured with EBI3-silenced melanoma cells.

Conclusion

This study filled the blank on the involvement of EBI3 in melanoma, hinting the possibility of controlling EBI3 as a therapeutic strategy in the management of melanoma.

Introduction

The Epstein-Barr virus induced gene 3 (EBI3) is a member of the IL-12 cytokine family, which includes four heterodimeric cytokines: IL-12 (p35/p40), IL-23 (p19/p40), IL-27 (p28/EBI3), and IL-35 (p35/EBI3). These cytokines utilize distinct combinations of five receptor chains: IL-12Rβ1, IL-12Rβ2, IL-23R, gp130, and WSX-1 for signaling. Unlike their disulfide-linked counterparts, IL-12 and IL-23, IL-27 and IL-35 do not form disulfide bonds, which results in poor pairing, leading to decreased stability and lower secretion levels. The roles of IL-12 and IL-23 as proinflammatory cytokines are well-established (Vignali & Kuchroo, 2012). In contrast, research involving IL-27 and IL-35 has been hindered by their instability in aqueous solutions and the scarcity of specific reagents (Vignali & Kuchroo, 2012). IL-27 has been suggested to have both proinflammatory and anti-inflammatory effects, as it can encourage Th1 polarization (Pflanz et al., 2002) while also inducing IL-10 production (Awasthi et al., 2007; Stumhofer et al., 2007). On the other hand, IL-35 is believed to exhibit anti-inflammatory actions, primarily by inhibiting T-cell proliferation and promoting the conversion of naive T cells into IL-10–producing T cells (Collison et al., 2007). There is speculation that EBI3 may be secreted and act as a homodimer, but this concept has yet to be clarified (Wirtz et al., 2005; Jensen et al., 2017). Moreover, an existing study has suggested that EBI3, which encodes a secretory glycoprotein of a molecular weight 34 kDa via a signal peptide and two fibronectin type III domains, is involved the carcinogenesis of Hodgkin’s lymphoma, adult T-cell lymphoma, and lung cancer, for instance (Nishino et al., 2011; Larousserie et al., 2005; Niedobitek et al., 2002). In the meantime, upregulated EBI3 expression level was revealed to share correlation with the adverse clinical outcome and malignant progression in breast cancer patients (Jiang & Liu, 2018). Further, downregulation of EBI3 was seen in cervical cancer tissues following radiotherapy, and the overexpression of EBI3 could play a preventive effect against radiation-induced immunosuppression on cervical cancer cells (Zhang et al., 2017). These evidences indeed proved the involvement of EBI3 in diverse cancers; however, the specific involvement of EBI3 in melanoma has not been explored, except a preliminary study based on The Cancer Genome Atlas (TCGA) hinting EBI3 as a potential biomarker in metastatic melanoma (Yonekura, 2022).

Melanoma refers to a skin cancer resulting from the malignancy of melanocytes with a rapidly growing incidence around the world (Ahmed, Qadir & Ghafoor, 2020; Grossi et al., 2023). The mortality associated with melanoma has risen along with the increasing incidence rates over the years, culminating in a statistic where one in every four deaths is attributed to this disease (Rastrelli et al., 2014). However, the treatment options for unresectable stage III and IV melanoma have undergone a significant transformation due to the advent of immunotherapies and targeted therapy approaches. These two methods have demonstrated considerably enhanced survival rates when compared to traditional chemotherapy regimens (Michielin et al., 2019). Since the US Food and Drug Administration (FDA) granted approval for ipilimumab in 2011—marking the introduction of the first immune checkpoint inhibitor to enhance survival in advanced cases (Teixido et al., 2021; Hodi et al., 2010)—and vemurafenib, which is the first in its class as a BRAF tyrosine kinase inhibitor derived from the v-raf murine sarcoma viral oncogene homolog B1, there has been a notable decline in melanoma mortality (Sosman et al., 2012; Chapman et al., 2011). Also, some repressive mechanisms can be seen in melanoma, which, in general, act together to escape both innate and adaptive immune detection and destruction (Marzagalli, Ebelt & Manuel, 2019). Hence, discovery on reliable biomarkers for the prediction of response to immunotherapy for melanoma patients remain an urgent need (Zeng et al., 2023).

In our current work, we further explored the effects of EBI3 in the malignant phenotype of melanoma cells and on the tumor microenvironment. Toward this end, the expression pattern of EBI3 in melanoma was predicted and the effects of EBI3 on the malignant phenotypes and immune of melanoma cells were investigated based on some basic cellular assays. Here, relevant results have demonstrated that EBI3 was highly-expressed in melanoma cells and silencing of EBI3 could modulate the malignant behaviors and immune response in melanoma. These results thus further completed the studies showing the implication of EBI3 in melanoma.

Materials and Methods

Cell culture

Human melanoma cell lines M14 (BNCC340599; BeNa Culture Collection, Beijing, China), A2058 (CL-0652; Procell, Wuhan, China), HMY1 (CL-0696; Procell, Wuhan, China), MV3 (BNCC339913; BeNa Culture Collection, Beijing, China) and A875 (BNCC359299; BeNa Culture Collection, Beijing, China) were applied for the quantification on EBI3 mRNA expression. Leibovitz’s L-15 medium (SH30525; Cytiva Lifesciences, Marlborough, MA, USA) or high-glucose Dulbecco’s modified Eagle’s medium (SH30022; Cytiva Lifesciences, Marlborough, MA, USA) with the supplementation of 10% bovine calf serum (SH30071; Cytiva Lifesciences, Marlborough, MA, USA) and 1% penicillin-streptomycin (SV30010; Cytiva Lifesciences, Marlborough, MA, USA). Human monocyte cell line THP-1 (BNCC358410; BeNa Culture Collection, Beijing, China) were additionally cultured in Roswell Park Memorial Institute-1640 medium (SH30255; Cytiva Lifesciences, Marlborough, MA, USA) plus 10% bovine calf serum, 1% penicillin-streptomycin and 0.05 mM β-mercaptoethanol (M301574; Aladdin, Shanghai, China). All cell lines were tested via STR method and free of mycoplasma contamination, and incubated in an incubator at 37 °C with 5% CO2. For the induction into macrophages in vitro, THP-1 monocytes were induced with 100 ng/mL phorbol 12-myristate 13-acetate (PMA, P408905; Aladdin, Shanghai, China) for 48 h, followed by the incubation in Roswell Park Memorial Institute-1640 medium (Phuangbubpha et al., 2023).

Cell transfection

The small interfering RNAs against EBI3 with specific target sequences (si EBI3-1 and si EBI3-2) as well as the control small interfering RNAs with random sequence were all synthesized and purchased from GenePharma (Shanghai, China, https://www.genepharma.com/), which were then transfected into melanoma cells with Lipofectamine 3000 transfection reagent (L3000-001; Thermo Fisher Scientific, Waltham, MA, USA) for 48 h at 37 °C as per the manuals. All cells were harvested and the transfection efficiency was tested via reverse-transcription quantitative PCR (RT-qPCR). The sequences for the transfection were listed in Table 1.

Table 1 Sequences for transfection.

Gene	Target sequence (5′-3′)	
si EBI3-1	CAGTTTCTCTAGCTGAGAAATGG	
si EBI3-2	GAGAAATGGAGATGTACTACTCT	
si NC	AGAGAAATGCTAGCTGTTTCTCG	

Cell proliferation test

Melanoma cells were counted and distributed in 96-well culture plates (2 × 103/well) for the culture at the indicated time points. Then, the culture medium for melanoma cells was replaced with the one containing 10 μL CCK-8 solution (CK04; Dojindo Laboratories, Kumamoto, Japan). Melanoma cells were then continued for 4-h incubation in the incubator at 37 °C with 5% CO2. The optical density (OD) value at the 450 nm was recorded in SPECTRO Star Nano microplate reader (BMG Labtech, Gary, NC, USA) and the viability of cells in each group was tested and plotted in a curve (Tian et al., 2023).

Cell migration/invasion test

Following the transfection for 48 h, 100 μL melanoma cells were harvested and inoculated into 24-well plates in the upper Transwell chamber (pore: 8 μm, 3422; Corning, Inc., Corning, NY, USA) coated with/out the Matrigel® matrix (354230; Corning, Inc., Corning, NY, USA), which was added with 200 μL serum-free cell culture medium. In the meantime, 600 μL complete medium was added to the lower Transwell chamber. A total of 48 h after the culture, the migrated and invaded cells were fixed and dyed, followed by the quantification under an optical microscope (Nikon, Tokyo, Japan).

ELISA

IL-35 has a significant immunosuppressive effect and is able to modulate the function of T cells and macrophages (Ye et al., 2021). And two cytokines, IL-4 and IL-13, which are closely related to the polarization of M2-type macrophages, are able to promote the formation of an immunosuppressive microenvironment, which in turn supports tumor growth and metastasis (Shi et al., 2021; Bernstein et al., 2023). To be able to assess the role of EBI3 in regulating the immune microenvironment of melanoma, we chose IL-35, IL-4, and IL-13 as indicators for assessing cellular immune response. In this study, melanoma cells were cultured in the six-well culture plates at the density of 8 × 105/well for 24 h. Hereafter, the cell culture supernatant was harvested for the calculation on the levels of relevant cytokines including IL-35 (NBP3-06774), IL-4 (NBP1-91171) and IL-13 (NBP1-91176) as per the manuals of the producer (Novus Biologicals, Centennial, CO, USA). The absorbance at 450 nm was read in the microplate reader.

Co-culture system preparation

Transwell chambers (pore: 0.4 μm, 3470; Corning, Inc., Corning, NY, USA) were applied and settled in a 24-well plate. Hereafter, 500 μL induced macrophages (3 × 105/mL) were transferred to the upper chambers, while the lower chambers were filled with 1 mL melanoma cells (2 × 105/mL). These plates were then put in the incubator for 8-h culture, as described elsewhere (Yan et al., 2022).

Cell immunofluorescence assay

Following the co-culture of melanoma and macrophages, the polarization of macrophages was tested in cell immunofluorescence assay. In detail, 4% fixative paraformaldehyde solution (P395744; Aladdin, Shanghai, China) and 0.5% Triton X-100 (T109026; Aladdin, Shanghai, China) were serially applied to fix and permeabilize the co-cultured cells, which were hereafter washed in PBS thrice (5 min for each time). Following the blocking in 5% goat serum (C0265; Beyotime Institute, Shanghai, China), cells were incubated with the following primary antibodies (Abs) against CD11b (Alexa Flour® 488, 53-0112-82; Thermo Fisher Scientific, Waltham, MA, USA), CD86 (Alexa Flour® 647, 105019; BioLegends, Inc., San Diego, CA, USA) and CD206 (Alexa Flour® 647, MA5-44147; Thermo Fisher Scientific, Waltham, MA, USA) overnight at 4 °C. After the washing in cold PBS twice, co-cultured cells were incubated in goat-anti rabbit IgG secondary antibody (ab6702; Abcam, Cambridge, UK) at ambient temperature for 1 h in a dark room. Following the staining of the nuclei with 4′,6-diamidino-2-phenylindole (DAPI, 62247; Thermo Fisher Scientific, Waltham, MA, USA), images of cells were captured in a fluorescence microscope (Nikon, Tokyo, Japan).

RNA extraction and reverse transcription-quantitative polymerase chain reaction

TRIzol reagent (15596026; Thermo Fisher Scientific, Waltham, MA, USA) was applied for the extraction of total cellular RNA. A total of 1 μg of the extracted RNA was reversely transcribed into complementary DNA (cDNA) using iScript™ cDNA synthesis kit (4106228; Bio-Rad, Hercules, CA, USA) as per the manuals. The qPCR was then carried out using iQ™ SYBR® green supermix (1708880; Bio-Rad, Hercules, CA, USA) in CFX96 touch real-time PCR detection system (Bio-Rad, Hercules, CA, USA) at the following cycling steps: 95 °C for 2 min and 40 repeated cycles of 95 °C for 15 s, 60 °C for 30 s and 72 °C for 30 s. The primers applied are listed in Table 2. 2−ΔΔCT method was applied for the calculation of the relative mRNA expressions with GAPDH as the housekeeping control (Livak & Schmittgen, 2001; Amuthalakshmi, Sindhuja & Nalini, 2022).

Table 2 Primer sequences applied in this study.

Gene	Accession no	Forward primer	Reverse primer	
EBI3	NM_005755.3	CTGGATCCGTTACAAGCGTCAG	CACTTGGACGTAGTACCTGGCT	
BAX	NM_138761.4	GAACTGATCAGAACCATCAT	ATCTTCTTCCAGATGGTGA	
BCL2	NM_000633.3	GGATGACTGAGTACCTGAAC	TGAGCAGAGTCTTCAGAGA	
CDH1	NM_004360.5	AGCCCTAATCATAGCTACAG	GTGGTCACTTGGTCTTTATT	
CDH2	NM_001792.5	TCCTGAAGATGTTTACAGTG	AAGAACTCAGGTCTGTTGTC	
GAPDH	NM_002046.7	CCTCAACTACATGGTTTACA	TGTTGTCATACTTCTCATGG	

Protein extraction and western blotting analysis

RIPA lysis buffer (P0013B; Beyotime Institute, Shanghai, China) was adopted to lyse the total cellular protein from melanoma cells or induced macrophages containing the protease and phosphatase inhibitor cocktail (P1045; Beyotime Institute, Shanghai, China) and the equal amount of protein sample was loaded into and detached in sodium dodecyl sulfate-polyacrylamide gels for electrophoresis. Subsequently, the protein samples were transferred to the polyvinylidene fluoride membranes (FFP32; Beyotime Institute, Shanghai, China) (Zhang et al., 2023), which were then blocked in 5% western blotting buffer (P0252; Beyotime Institute, Shanghai, China) and incubated with the following primary Abs (Abcam, Cambridge, UK) against p-SMAD3 (ab74062, 1/1,000 dilution), SMAD3 (ab40854, 1/1,000 dilution), signal transducer and activator of transcription 6 (STAT6, ab32108, 1/5,000 dilution), p-STAT6 (ab263947, 1/1,000 dilution), cyclic GMP-AMP synthase (c-GAS, ab302617, 1/1,000 dilution), p-stimulator of interferon genes (STING, ab318181, 1/1,000 dilution), STING (ab239074, 1/1,000 dilution), p-inositol-requiring enzyme 1α (IRE1α, ab124945, 1/2,000 dilution), IRE1α (ab96481, 1/2,000 dilution) and housekeeping control GAPDH (ab8245, 1/10,000 dilution) at 4 °C overnight. Further, the membranes were reacted with horseradish peroxide-conjugated secondary Abs against rabbit IgG (ab205718; Abcam, Cambridge, UK) or mouse IgG (ab205719; Abcam, Cambridge, UK) at 1/5,000 dilution for 1-h incubation at ambient temperature.

Finally, for the visualization process, the membranes were sequentially rinsed in TBST buffer (ST671; Beyotime Institute, Shanghai, China) and exposed in BeyoECL Plus visualization reagent (P0018S; Beyotime Institute, Shanghai, China). The analysis was implemented in ChemiDoc™ touch imaging system (Bio-Rad, Hercules, CA, USA) and the densitometry on the grey value was carried out with the help of ImageJ 5.0 (Bio-Rad, Hercules, CA, USA).

Statistics

All statistical analyses were implemented applying GraphPad Prism 7 (GraphPad, Inc., La Jolla, CA, USA) and all data were compared with student’s t-test and presented as mean ± standard deviation. Changes at the significance level of p-value < 0.05 were assumed to be statistically significant.

Results

Pan-cancer EBI3 expression analysis

With the purpose of exploring the involvement of EBI3 in melanoma, the pan-cancer EBI3 expression pattern was predicted and downloaded (Fig. 1A). Notably, skin cutaneous melanoma was screened out and the level of EBI3 was additionally predicted, and an elevated EBI3 expression was observed (Fig. 1B, p-value < 0.05). Therefore, in our subsequent assays, the corresponding small interfering RNAs with EBI3-specific targeting sequences were applied for the knockdown assays.

Figure 1 Pan-cancer EBI3 expression analysis.

(A) Pan-cancer EBI3 expression analysis based on the data from TCGA. (B) Expression analysis on EBI3 level in skin cutaneous melanoma. SKCM, Skin cutaneous melanoma.

Effects of EBI3 knockdown on the malignant phenotypes of melanoma cells

Subsequently, the expression levels of EBI3 in melanoma cells were quantified in the beginning, and the highest expression of EBI3 was seen in MV3 cells (Fig. 2A, p-value < 0.01). Then the relevant small interfering RNAs with EBI3-specific targeting sequences were constructed and transfected into MV3 cells via liposome, and the successful knockdown of EBI3, as seen by the decreased EBI3 mRNA expression (Fig. 2B, p-value < 0.001). The relevant results from the Cell Counting Kit-8 (CCK-8) assay then suggested that silencing of EBI3 using these two specific small interfering RNAs could visibly diminish the viability of MV3 cells (Fig. 2C, p-value < 0.01).

Figure 2 Effects of EBI3 knockdown on the viability of melanoma cells.

(A) The mRNA expression analysis on EBI3 in melanoma cell lines (M14, A2058, HMY1, MV3, and A875). (B) Validation on the transfection of EBI3-specific small interfering RNAs in melanoma cells via RT-qPCR. (C) Effects of EBI3-specific small interfering RNAs on the viability of melanoma cells via CCK-8 assay. ns: p > 0.05; *p < 0.05; **p < 0.01; ***p < 0.001. SKCM, Skin cutaneous melanoma.

Transwell migration/invasion assay was adopted to evaluate the migration and invasion of melanoma cells following the knockdown of EBI3. As shown in Figs. 3A and 3B, at 48 h, the reduced number of both migrated and invaded melanoma cells was clearly seen following the knockdown of EBI3 (Figs. 3A and 3B, p-value < 0.01). Also, to further explore the potential effects of EBI3 on epithelial-mesenchymal transition (EMT) signatures, we examined the mRNA expression levels of CDH1 and CDH2 upon EBI3 silencing. As shown in Fig. 3C, the silencing of EBI3 could contribute to the reduced expression of CDH2 yet the increased level of CDH1 (p-value < 0.05). Further, the expression levels of the apoptosis-related regulator BCL2 family proteins (BAX and BCL2) were quantified, and the silencing of EBI3 led to the suppressed BCL2 expression and the enhanced BAX expression in MV3 cells (Fig. 3D, p-value < 0.05). These findings further support the critical role of EBI3 in promoting the malignant behavior of melanoma; not only does EBI3 play a facilitating role in tumor cell migration, invasion, and survival, but its knockdown not only inhibits the EMT process but also induces apoptosis.

Figure 3 Effects of EBI3 silencing on the migration, invasion and apoptosis modulators in melanoma cells.

(A and B) Transwell migration/invasion assay evaluating the migration and invasion of EBI3-silenced melanoma cells MV3 at 48 h. (C) Relative mRNA levels of metastasis-related cadherins (CDH1 and CDH2) in EBI3-silenced melanoma cells MV3. (D) Relative mRNA levels of apoptosis-related genes BAX and BCL2 in EBI3-silenced melanoma cells MV3. *p < 0.05; **p < 0.01.

Effects of EBI3 knockdown on the phosphorylation of SMAD3 and STAT6 in melanoma cells

The activation of two pathways, TGF-β and IL-4/IL-13, play important roles in malignant behaviors such as tumor cell growth, migration, and immune response; therefore, by investigating the phosphorylation levels of Smad3 and STAT6 (Zeng et al., 2024; Hu et al., 2024), it is possible to further understand whether EBI3 can have an effect on the presence of melanoma through these pathways. In this study, we focused on Smad3 and STAT6 in melanoma cells, and the phosphorylation of these two proteins was determined in western blotting assay accordingly. The relevant quantification results in this study demonstrated that the phosphorylation of both Smad3 and STAT6 was obviously diminished by EBI3-specific small interfering RNA (Figs. 4A–4C, p-value < 0.05).

Figure 4 Effects of EBI3 knockdown on the phosphorylation of SMAD3 and STAT6 as well as immune response in melanoma cells.

(A–C) Quantification on the phosphorylation of SMAD3 and STAT6 in EBI3-silenced melanoma cells MV3 via western blotting. (D) Measurement on the immune response-related cytokines IL-35, IL-4 and IL-13 in EBI3-silenced melanoma cells MV3 via ELISA. *p < 0.05; **p < 0.01.

Effects of EBI3 knockdown on the immune response in melanoma cells

ELISA was used to investigate the potential effects of EBI3 knockdown on the immune response of melanoma cells via calculating the concentrations of specific cytokines. The relevant results have demonstrated that the concentrations of these specific cytokines including IL-35, IL-4 and IL-13 were all higher in melanoma cells with the presence of EBI3-specific small interfering RNA (Fig. 4D, p-value < 0.05).

Effects of EBI3 knockdown on the macrophage polarization

Additionally, the effects of EBI3 knockdown on the macrophage polarization were further explored, and the scheme for immunofluorescence assay was listed in Fig. 5A. Then the corresponding results from immunofluorescence assay has suggested that the knockdown of EBI3 in melanoma cells contributed to the reduced CD206 fluorescence intensity yet the increased CD86 fluorescence intensity in these induced macrophages from THP-1 monocytes (Figs. 5B–5E, p-value < 0.05).

Figure 5 Effects of EBI3 knockdown on the macrophage polarization.

(A) Scheme for immunofluorescence assay. (B and C) Mean fluorescence intensity of CD86 in these co-cultured cells. (D and E) Mean fluorescence intensity of CD206 in these co-cultured cells. *p < 0.05.

The levels of relevant macrophage polarization indicators have been further quantified via western blotting. It was seen in THP-1 monocytes-induced macrophages that the silencing of EBI3 in melanoma cells could visibly upregulate the expression of c-GAS and promote the phosphorylation of both STING and IRE1α (Figs. 6A and 6B, p-value < 0.05). This suggests that knockdown of EBI3 activates the cGAS-STING and IRE1α signaling pathways in THP-1-derived macrophages, which may be associated with enhanced immune response and pro-inflammatory status and contribute to the polarization of M1-type macrophages.

Figure 6 Effects of EBI3 knockdown on the macrophage polarization-related markers.

(A and B) The protein expression or the phosphorylation level of macrophage polarization-related markers (STING, c-GAS and IRE1α in THP-1 monocytes-derived macrophages following the co-culture with EBI3-silenced melanoma cells. *p < 0.05.

Discussion

EBI3, a 34 kDa glycoprotein containing a signal peptide and two fibronectin type III domains yet lacking a membrane-anchoring motif, was previously known as one of the β-subunit of the IL-12 family of cytokines, which is secreted and present on the plasma membrane and endoplasmic reticulum (Watanabe et al., 2021). Previous studies have laid great emphasis on the association between EBI3 expression level and the prognosis of specific cancers like breast cancer (Jiang & Liu, 2018), acute myeloid leukemia (Wu et al., 2022) and cervical cancer (Hou et al., 2016), and some experimental analysis have further contributed to the understanding on the participation of EBI3 in colorectal cancer (Liang et al., 2016). In our current study, we further explored the involvement of EBI3 in melanoma based on some preliminary investigations using melanoma cells cultured in vitro. Relevant results have demonstrated that EBI3 was higher expressed in melanoma and the silencing of EBI3 using commercial small interfering RNAs could suppress the malignant phenotypes of melanoma cells as well as modulate the macrophage polarization.

While exploring the specific malignant phenotypes of EBI3-silenced melanoma cells in vitro, a series of assays like CCK-8 and Transwell were adopted, revealing that the silencing of EBI3 could suppress the proliferation and metastasis of melanoma cells. Similar trends were similar to those in colorectal cancer, where EBI3 blocking peptide could repress the proliferation of colorectal cancer and tumor growth (Liang et al., 2016). Our current study has hinted that EBI3 silencing could suppress the proliferation, migration and invasion of melanoma cells, concurrent with the diminished expression of CDH2 and BCL2 yet the upregulated expression of CDH1 and BAX. Cadherins are those crucial modulators which play a crucial role in tissue homeostasis and EMT refers to a mechanism characterized by the loss of E-cadherin (CDH1) and the acquisition of N-cadherin (CDH2) (Kaszak et al., 2020; Meng et al., 2024). Additionally, transition of normal melanocytic cells to malignant melanoma has a characteristic of EMT, which includes the disrupted adherens junctions resulted from the down-regulation of CDH1 and the up-regulation of CDH2 (Hao et al., 2012). BCL2 proteins are those crucial modulators which enclose some pro- and anti-apoptotic factors like BCL2 and BAX and participate in the survival and apoptosis of melanoma (Eberle & Hossini, 2008; Chipuk, 2015). The modulation of phenotypes as well as these proteins belonging to cadherin and BCL2 families via EBI3 has further completed the discovery highlighting the involvement of EBI3 in melanoma.

Cancer-related inflammation is present at different stages during tumorigenesis, causing genomic instability, epigenetic modification as well as the induction on the malignant phenotypes of cancer cells (Hanahan & Weinberg, 2011). Some soluble factors like cytokines have been revealed to be involved in this process (Christofides et al., 2022). IL-35 is a member of the IL-12 family with anti-inflammatory process, while IL-4 and IL-13 are two structurally and functionally related cytokines to T help 2 cells which can modulate the immune response and tumor microenvironment under normal physiological conditions and cancer (Xue, Yan & Kan, 2019; Suzuki et al., 2015). While trying to link these cytokines with melanoma, some published studies have suggested that IL-4 could trigger the anti-tumor activity of natural killer cells in metastatic melanoma patients (Vuletić et al., 2020). Here, we expanded the discoveries concerning the cytokines in melanoma based on some relevant results in our current study, where it was evident that the levels of IL-4, IL-13 and IL-35 in melanoma were all elevated following the silencing of EBI3. Macrophages are those most extensively investigated pleiotropic cells of the immune system, which are required for cancer cell growth and metastasis and also revealed to be implicated in all stages of melanomagenesis (Feng, Zhang & Chen, 2024; Pieniazek, Matkowski & Donizy, 2018). Generally, macrophages can be classified into M1 or M2 activation phenotypes with distinct role in melanoma (Bardi, Smith & Hood, 2018). Bardi, Smith & Hood (2018) discovered that an increase in markers characteristic of both M1 and M2 polarization phenotypes featured CCL22, IL-12B, IL-1 beta, IL-6, i-NOS, and TNF-alpha. In addition, Azumi et al. (2023) found in melanoma that THGP promotes M1 polarization in macrophages and inhibits the expression of signal-regulated protein alpha (SIRP-alpha) and CD47 in macrophages. In our current study, the silencing of EBI3 in melanoma cells could weaken the mean fluorescence intensity of CD206 yet promote that of CD86 in induced macrophages, along with the elevated levels of cGAS and the phosphorylation of STING and IRE1α. Further, STAT6 phosphorylation is a crucial signaling event acting as the downstream of the IL-4 receptor complex, while Smad3 is a constituent of a relevant signaling in promoting melanoma (Gong et al., 2012; Li et al., 2018). These findings suggest that inhibition of EBI3 may enhance anti-tumor immune responses through activation of immune-related signaling pathways, offering potential for melanoma treatment.

Nonetheless, there are some limitations to our study. First, this study explored the effects of EBI3 on the malignant phenotype and immune response of melanoma only at the cellular level. In the future, a mouse xenograft model will be introduced to verify the role of EBI3 on tumor growth, metastasis, and immune regulation in vivo. Second, we did not evaluate the regulatory role of other immune cell types besides macrophages, which prompts us to further analyze other potential roles of EBI3 in the future to reveal its combined impact in regulating immune escape. Finally, although this study observed changes in IL-35, IL-4, and IL-13 levels after knockdown of EBI3, the specific role of these cytokines in melanoma was not analyzed in detail. The specific mechanisms of action of these cytokines will be further explored in subsequent studies to confirm their specific contributions in the regulation of melanoma progression by EBI3.

In summary, the present study revealed the critical role of EBI3 in melanoma. high expression of EBI3 promoted proliferation, migration and invasion of melanoma cells, while its knockdown inhibited these malignant behaviors and induced apoptosis. In addition, knockdown of EBI3 significantly activated Smad3 and STAT6 signaling inhibition, enhanced the secretion of cytokines such as IL-35, IL-4, and IL-13, and activated cGAS-STING and IRE1α signaling in macrophages to promote M1-type polarization. This study provides a new rationale for EBI3 as a potential therapeutic target for melanoma.

Supplemental Information

Supplemental Information 1 MIQE checklist.

Abbreviations

EBI3 Epstein-Barr Virus Induced Gene 3

IL interleukin

TCGA the cancer genome atlas

PMA phorbol 12-myristate 13-acetate

RT-qPCR reverse-transcription quantitative PCR

OD optical density

Abs antibodies

DAPI 4′,6-diamidino-2-phenylindole

cDNA complementary DNA

STAT6 signal transducer and activator of transcription 6

c-GAS cyclic GMP-AMP synthase

STING stimulator of interferon genes

IRE1α inositol-requiring enzyme 1α

EMT epithelial-mesenchymal transition

METTL14 Methyltransferase Like 14

STAT3 signal transducer and activator of transcription 3

Additional Information and Declarations

Competing Interests

Author Contributions

Data Availability

The authors declare that they have no competing interests.

Ying Zhang conceived and designed the experiments, analyzed the data, prepared figures and/or tables, authored or reviewed drafts of the article, and approved the final draft.

Fengrui Cheng performed the experiments, prepared figures and/or tables, and approved the final draft.

Xingrui Cai conceived and designed the experiments, authored or reviewed drafts of the article, and approved the final draft.

Jingping Wu performed the experiments, analyzed the data, authored or reviewed drafts of the article, and approved the final draft.

The following information was supplied regarding data availability:

The experimental raw data is available at GitHub and Zenodo:

- https://github.com/JingpingWu637/Experimental-raw-data.git.

- JingpingWu637. (2024). JingpingWu637/Experimental-raw-data: Experimental raw data (v.1.1.0). Zenodo. https://doi.org/10.5281/zenodo.13841191.

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
