# Peer review of "Malignant behaviors and immune response in melanoma: Epstein-Barr virus induced gene 3 as a therapeutic target based on an in-vitro exploration"

_PeerJ, doi:10.7717/peerj.18730_

## Round 0.1 · original submission · Major Revisions

In the Introduction:
1. Expand the background on EBI3 beyond basic description of its localization and secretion.
2. Systematically explain why EBI3 has drawn attention in various cancers (leukemia, cervical cancer, breast cancer)
3. Add information about current clinical treatments for melanoma, including their advantages and disadvantages
In Materials and Methods:
1. Streamline the protein extraction and Western blot section to focus on key methodological details
2. Explain the rationale for selecting specific immune response indicators
3. Add details about immune checkpoint molecules studied
In Results:
1. Define all abbreviations used in Figures 1a and 2a in the figure legends
2. Clarify the time unit in Figure 2c (hours vs minutes)
3. Explain the calculation method for migrating/invading cells in Figure 3
4. Describe what EBI3 silencing implies for the EMT process (Figure 3c)
5. Provide rationale for studying Smad3 and STAT6 phosphorylation
6. Expand the description of Figure 6a results
In Discussion:
1. Add information about the role of macrophage polarization phenotype in melanoma progression
2. Discuss which cytokines and inflammatory factors are involved
3. Address why only macrophage polarization was studied among immune cells

Reviewer 1 ·

Basic reporting

(1) English Proficiency Level: The text could benefit from further refinement; sentences tend to be verbose. It's recommended to vary sentence length for better readability. For instance, "The Epstein-Barr Virus Induced Gene 3 (EBI3) is part of the interleukin (IL)-12 family, which is structurally similar to the p40 subunit. EBI3 can pair with either the IL-27p28 subunit or the IL-12p35 subunit to form heterodimers, leading to the creation of IL-27 or IL-35."
(2) Accuracy of References: The references are accurate.
(3) Content: Overall, the paper shows significant innovation. The experimental approach is conventional, and the study's depth is limited.

Experimental design

(1) Ethics: All cell lines have been tested for mycoplasma contamination and undergone STR profiling.
(2) Introduction:
To enhance the paper's originality, the background section should be enriched with the latest advancements in EBI3 research.

Validity of the findings

(1) Innovation: Current research on the role of EBI3 in melanoma is limited, which makes this paper innovative.

(2) Data Reliability:
1) In figures 1a and 2a, abbreviations are used. Please define these abbreviations in the figure captions.
2) The time unit in figure 2c needs clarification. Is the 12 representing minutes or hours?
3) The statistical results in figure 3b do not seem to align with figure 3a. Please provide a detailed explanation of how the number of migrating and invading cells was calculated.
4) Please briefly describe the results of figure 3c, indicating what the silencing of EBI3 implies for the EMT process.
5) The statement "These discoveries hence suggested that silencing of EBI3 could suppress the 195 proliferation and metastasis of melanoma cells in vitro" is incomplete and does not encompass all findings.
6) What is the rationale for studying the phosphorylation levels of Smad3 and STAT6? Please explain this in the text.
7) What do the results of figure 6a demonstrate? Please describe this in the text.
(3) Conclusion: Please present a standalone conclusion section following the discussion and summarize the findings of this paper.

Additional comments

(1) Image Resolution: Please provide high-resolution images where text is legible and cell details are clearly visible.
(2) Abstract: The background section should succinctly outline the context and objectives of this paper. Please revise accordingly. The methods section should refrain from stating objectives and instead summarize the methods used.

Reviewer 2 ·

Basic reporting

no comment

Experimental design

no comment

Validity of the findings

no comment

Additional comments

Taking the regulatory role of EBI3 on melanoma as an entry point, this study aims to investigate the basic situation and related mechanisms of EBI3 involvement in the malignant behavior and immune response of melanoma. This study mainly quantified the level of EBI3 in melanoma cells by molecular assays, and assessed the regulatory role of EBI3 on the survival and migration of melanoma cells by a series of cellular experiments such as CCK-8 and Transwell. Next, this study took a cell co-culture approach to explore the tangible regulatory role of EBI3 in melanoma cells on macrophage polarization phenotype and to elucidate the regulatory role of this gene on immune response-related cytokines. In conclusion, this study offers the possibility of revealing EBI3 as a therapeutic strategy for melanoma and is overall eligible for publication, but the following issues still need to be addressed before publication:
1. The abstract section of this study needs further refinement, such as elucidating the molecular assays by which the expression levels of EBI3 in melanoma were revealed, and elucidating the means by which the EBI3 gene was knocked down.
2. The theme of this study was to elucidate the role of EBI3 in the regulation of malignant behavior and immune responses in melanoma, but why in the actual study was the focus only on exploring the role of this gene in the regulation of macrophage polarization phenotypes and did not investigate its relationship with other immune cells? Please explain.
3. Does this study have any rationale for the selection of indicators related to immune response and why were these indicators selected as indicators for assessing cellular immune response? Please explain and add to the Materials and Methods.
4. Is there any basis for the study's claim that it fills a gap in melanoma research? Is it possible that existing studies have not explored the role of this gene in melanoma? Please provide the basis for this.
5. This study demonstrates that standard treatments for melanoma have not substantially improved the clinical outcome of melanoma, so what are the more common and effective clinical treatments available? What are the advantages and disadvantages of each of these treatments? Please provide additional information.
6. Since this study investigates the role of genes in the regulation of immune escape in melanoma, which immune checkpoint molecules were selected for focus in this study? This does not appear to be reflected in the original article, and it is suggested that this be added and, if necessary, the relevant statement in the introduction regarding “immune escape” be revised.
7. The section on protein extraction and WB experiments in Materials and Methods is too lengthy, and it is recommended that this be streamlined by focusing on what reagents, instruments and incubation conditions are used.
8. The description of EBI3 in this study is too basic, only indicating the localization and secretion function of the gene, but not systematically clarifying why it is closely associated with leukemia, cervical cancer, breast cancer, etc., or why it has attracted much attention in the medical field. It is therefore recommended that additional information be provided and reflected in the introduction and discussion.
9. It is recommended that additional reports in the literature be added to illustrate the important regulatory role of the macrophage polarization phenotype in melanoma for disease progression, in particular to reveal which cytokines and inflammatory factors are involved in the release of this regulatory process.
10. It is recommended that additional information on the limitations of this study be added and specified in this regard. At the same time, the experimental ideas for the follow-up of this study could be further elucidated, such as whether further tissue experiments will be conducted to validate the initial ideas of this study.

---

## Round 0.2 · accepted · Accept

This manuscript has been significantly improved after authors' revisions. All concerns have been addressed.

Therefore, I think this paper can be accepted for publication.

Reviewer 1 ·

Basic reporting

Thank you for the author's point-to-point response. I no longer have any new comments.

Experimental design

no comment

Validity of the findings

no comment

Reviewer 2 ·

Basic reporting

This study mainly quantified the level of EBI3 in melanoma cells through molecular detection, and evaluated the regulatory effect of EBI3 on melanoma cell survival and migration through a series of cell experiments such as CCK-8 and Transwell. Next, this study used cell co culture method to explore the tangible regulatory effect of EBI3 on macrophage polarization phenotype in melanoma cells, and elucidate the regulatory effect of this gene on immune response related cytokines. In summary, this study provides the possibility of revealing EBI3 as a treatment strategy for melanoma, starting from the regulatory effect of EBI3 on melanoma, and exploring the basic situation and related mechanisms of EBI3's involvement in melanoma malignant behavior and immune response.

Experimental design

no comment

Validity of the findings

no comment